# Environmental and molecular noise buffering by the cyanobacterial clock in individual cells

Aleksandra Eremina[1], Christian Schwall[1], Teresa Saez[1], Lennart Witting [2], Dietrich Kohlheyer [2], Bruno M. C. Martins [3] ✉, Philipp Thomas [4] ✉ & James C. W. Locke [1] ✉

Circadian clocks enable organisms to anticipate daily cycles, while being robust to molecular and environmental noise. Here, we show how the clock of the cyanobacterium *Synechococcus elongatus PCC 7942* buffers genetic and environmental perturbations through its core KaiABC phosphorylation loop. We first characterise single-cell clock dynamics in clock mutants using a microfluidics device that allows precise control of the microenvironment. We find that known clock regulators are dispensable for clock robustness, whilst perturbations of the core clock reveal that the wild type operates at a noise optimum that we can reproduce in a stochastic model of just the core phosphorylation loop. We then examine how the clock responds to noisy environments, including natural light conditions. The model accurately predicts how the clock filters out environmental noise, including fast light fluctuations, to keep time while remaining responsive to environmental shifts. Our findings illustrate how a simple clock network can exhibit complex noise filtering properties, advancing our understanding of how biological circuits can perform accurately in natural environments.

Circadian clocks enable a diverse array of organisms to adapt to the periodic fluctuations in the environment caused by the Earth's rotation. To achieve this adaptation, clocks orchestrate 24 hour rhythms in key biological processes, such as metabolism and growth in prokaryotes[1,2] and plants[3,4], and the sleep/wake cycle in mammals[5]. For accurate timing of daily cycles, circadian systems must not only sense the environment and synchronise with it but also exhibit robustness to environmental perturbations. For instance, clocks should adjust to seasonal changes in the length of the day, but must avoid mistiming during a cloudy day. Clocks must also be robust to gene expression noise arising from low numbers of molecular components. While extensive experimental and theoretical work has explored noise buffering in clocks[6–12] and, separately, their ability to tune their response (plasticity) to entraining stimuli[13,14], the mechanisms governing the balance between robustness and plasticity in clocks are still unclear.

The cyanobacterium *Synechococcus elongatus* PCC 7942 is an excellent model for investigating the clock dynamics, due to the simple structure of its clock and the ease of imaging clock reporter expression in individual cells[15,16]. Its core clock network consists of just three genes, *kaiA*, *B*, and *C*, which generate a 24-h post-translational oscillation in KaiC protein phosphorylation[17,18]. The phospho-status of KaiC indirectly controls the expression of most of the cyanobacterial genome and affects such key processes as cell growth, division, and photosynthesis[19]. The cyanobacterial clock is entrained by light and temperature upon metabolic changes in quinone redox state[20] and ATP/ADP ratio in the cell[21], which influence the KaiC phosphorylation cycle. KaiABC-based oscillations can also be reconstructed in vitro[18,22], and there is no evidence of cell-to-cell coupling in vivo[16,23]. However, although the core clock is simple, multiple additional regulatory genes have been found to interact with the core clock network and tune clock

[1]Sainsbury Laboratory, University of Cambridge, Cambridge, UK. [2]IBG-1: Biotechnology, Forschungszentrum Jülich, Jülich, Germany. [3]University of Warwick, Warwick, UK. [4]Imperial College London, London, UK. ✉e-mail: bruno.martins@warwick.ac.uk; p.thomas@imperial.ac.uk; james.locke@slcu.cam.ac.uk

rhythms[24–28]. It is still not clear if these are necessary for clock robustness in the wild.

In order to study clock robustness, single-cell approaches are required to resolve system dynamics otherwise masked in bulk measurements[29]. Currently, the state-of-the-art methods to investigate cyanobacterial clock dynamics at such resolution are based on time-lapse microscopy using agarose-based environments[30,31]. Although a powerful and simple technique, it is becoming increasingly clear that agarose pads have limitations, such as environmental heterogeneity, imaging artefacts[32], and limited experiment duration, all of which could affect measurements of clock robustness. While attempts have been made to overcome these issues by micropatterning channels in agarose pads[33,34], controlled liquid micro-environments enabled by microfluidic devices, such as those developed for other bacteria, would advance our single-cell imaging capabilities. In this study, we present such a device, based on the mother-machine set-up[35], which we optimised for cyanobacterial growth.

Despite the limitations of available single-cell methods, previous assessments of noise in individual cyanobacterial oscillators have demonstrated the clock's remarkable robustness[16,23,36]. In constant environmental conditions, wild type (WT) cyanobacterial clocks exhibit an autocorrelation time of several weeks[16]. While the transcriptional feedback of KaiC onto its own promoter has been proposed to contribute to this robustness[34], the involvement of other regulators remains unclear. Notably, with a few exceptions[37,38], single-cell analyses of the clock have predominantly focused on free-running rhythms. Although the cyanobacterial clock has been shown to be responsive to light changes in bulk studies[39,40], the robustness of individual cyanobacterial clocks in responding to such perturbations remains unclear.

In this study, we develop a microfluidic growth system for cyanobacteria, ensuring a stable growth environment during long-term imaging. We first use this device to examine clock robustness under a range of constant light levels and genetic modifications to clock regulators, finding lower noise than previously estimated and confirming Aschoff's rule, where clocks of diurnal organisms run faster in higher light[41]. However, period- and rhythmicity-affecting mutations in KaiC significantly increase system noise. Combining experiments with a simple phosphorylation-based stochastic model of the core clock network, we find the clock buffers noise in light/dark switch timing, adjusting its phase proportionally to advances and delays in that timing. The clock also filters out transient fluctuations in light levels during the day under natural light/dark conditions, with only a small increase in cell-to-cell variability compared to square wave light/dark cycles. Our work sheds light on the robustness of the cyanobacterial clock under diverse perturbations and supports the idea that clock robustness can be understood through core network properties.

## Results

### The cyanobacterial clock is extremely robust over a range of constant light levels

To investigate the robustness of individual circadian oscillators, we optimised a mother machine microfluidic device[42] specifically for enhanced cyanobacterial cultivation. The connection between the top of the growth channel and the feeding channel in this device, which we found to be critical for robust cyanobacterial growth, allows improved cell loading and fluid flow in growth channels (Supplementary Fig. 1). We developed a protocol for multi-day growth and imaging of *S. elongatus*, which required modifications to the standard conditions used for *Escherichia coli* (Methods). We adapted and further developed an existing pipeline[43] to analyse the time-lapse fluorescence microscopy images we generated (Supplementary Fig. 2). We named this set-up the 'Green Mother Machine' (GMM). As demonstrated throughout this paper, we could precisely control the light environment in the GMM, reproducing light profiles of different complexity, from continuous light to realistic meteorological conditions.

To examine single-cell clock dynamics, we first imaged a chromosomally integrated reporter for the clock[44], *pkaiBC:eYFP-fsLVA* (Methods), under constant light conditions (Fig. 1a-b, see Supplementary Movie 1 for example data set) and extracted information about cell growth and clock dynamics (Fig. 1c-d). This same reporter has been used in previous single-cell clock studies[34,36,38], allowing direct comparison to their work (see Methods and Table 1 for details). We validated the setup by observing cell growth and division rates comparable to previous reports on agarose pads[2] (doubling time reported as 9.8 ± 2.3 h (mean ± standard deviation) at 25 μmol m$^{-2}$ s$^{-1}$, compared to 9.9 ± 3.6 h in our data at 20 μmol m$^{-2}$ s$^{-1}$). The free-running oscillations were highly synchronised, with sinusoidal waveforms similar to previous reports under comparable light conditions[2] (Fig. 1d). When investigating the free-running clock dynamics under three constant light conditions (LL, 10, 20, and 40 μmol m$^{-2}$ s$^{-1}$, Fig. 1d and Supplementary Fig. 3), we observed that the mean period decreases with the level of light, in accordance with Aschoff's rule[41] (Fig. 1e). Our single-cell observations confirm previous findings in bulk experiments[24]. However, the overlap of the period distributions under different LL levels suggests that individual lineages cannot be unequivocally mapped to specific conditions based just on their clock periods.

To further quantify the precision of individual oscillators, we measured the clock's phase diffusion time, which estimates the time it takes for fluctuations to randomise the clock phase, and its autocorrelation time, which estimates the decay of correlations in the reporter signal due to the clock's phase and amplitude fluctuations (Supplementary Fig. 4). The long timescales that we quantified imply that the free-running cyanobacterial clocks are exceptionally stable (Fig. 1f): $D$, the phase diffusion coefficient, was $\sim 9.0 \times 10^{-5} \pm 3.9 \times 10^{-5} h^{-1}$ under high LL (Methods), which is lower, and implies lower noise levels, than in previous reports[16] ($D = 5 \times 10^{-4} \pm 3 \times 10^{-4} h^{-1}$ under 100 μmol m$^{-2}$ s$^{-1s}$). In our microfluidic set-up, clocks diffused by only a few minutes every day:

$$\sigma(\Delta\varphi) = \sqrt{(D \times \Delta t)} \qquad (1)$$

where $\sigma(\Delta\varphi)$ is the standard deviation of phase differences and $\Delta t = 24\,h$, was 1–1.7 h. This resulted in timing errors of 5–7% in the free-running clock under a range of LL conditions used (compared to larger errors of 10–12% obtained in Mihalcescu et al.[16], although similar to the timing error observed in Chew et al.[45]). Similarly, previous estimates of clock robustness provided phase diffusion times of 60–260 days[16]. Phase diffusion times measured in our setup were around 750–1500 days and decreased to 250–750 days in low light conditions (Fig. 1f, left) demonstrating unprecedented levels of coherence. Estimates of correlation times obtained from autocorrelation functions were lower (Fig. 1f, right), suggesting that, besides phase diffusion, fluctuations in amplitude and downstream components (e.g. clock reporter) contribute to temporal correlations. We observe that despite the period plasticity (Fig. 1e), the rhythm stability in our data is high for all the LL intensities examined. This stability is also reflected in high synchronicity amongst the clock in different cell lineages, as illustrated in Fig. 1h (top panel). We observe similar drops in synchronicity in our medium and low light experiments as those previously measured using an agarose-based microfluidics setup[34].

Evaluating the relationship between gene expression variability (C.V.$^2$) and the level of gene expression through the circadian cycle reveals a noise loop (Fig. 1g, Supplementary Fig. 5). As described previously by Chabot et al.[36], noise levels in opposite phases of the circadian cycle are not the same. There are also two maxima in inter-lineage variability in gene expression for each clock period, as observed across the three LL conditions in our data (Supplementary Fig. 6). Although noise amplitude values are difficult to directly compare given differences in imaging setups and analyses, our expression noise measurements of C.V.$^2$ under 0.05 are markedly lower than the

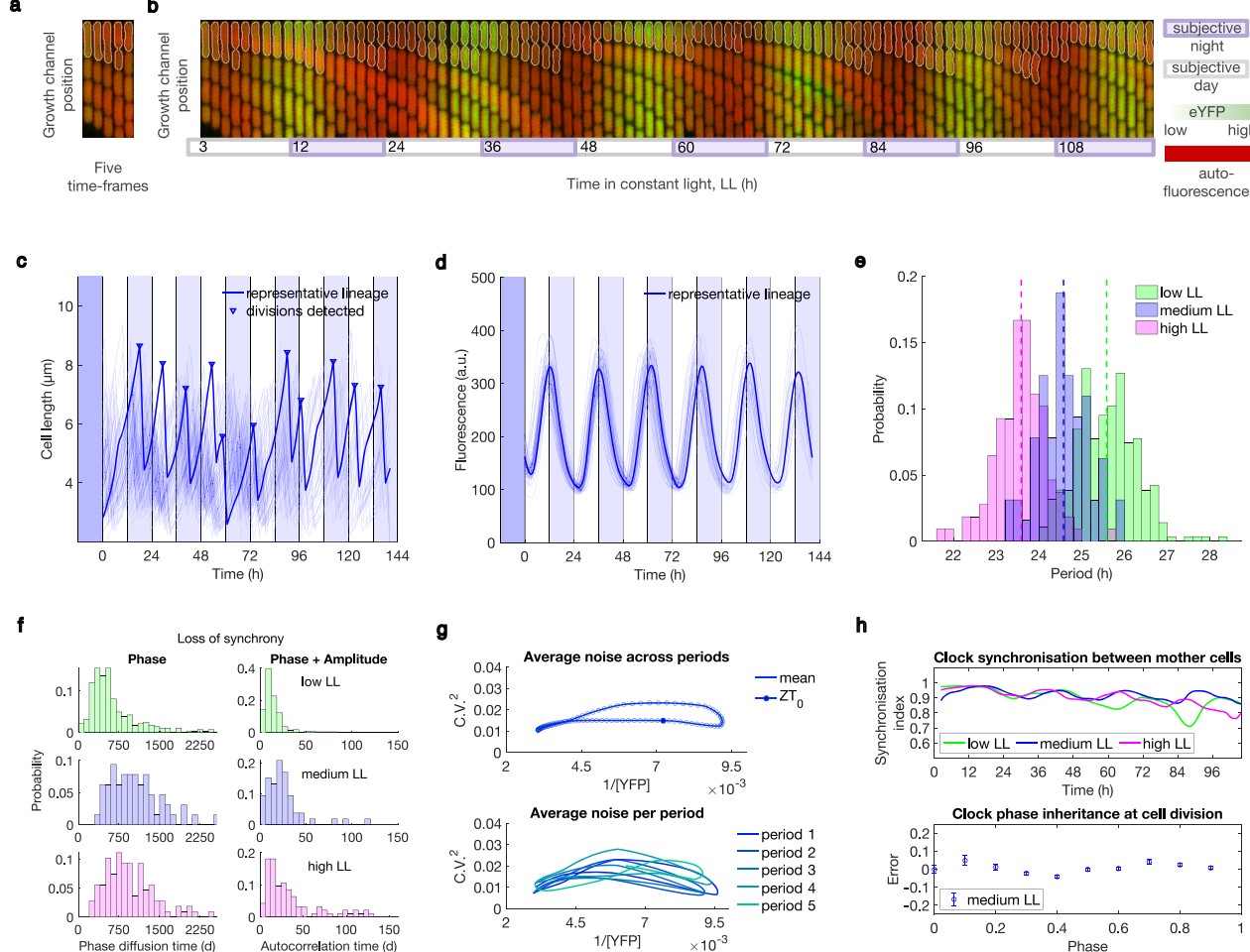

**Fig. 1 | Single-cell cyanobacteria have highly robust clocks across a range of constant light levels. a, b** Five time-frames (**a**) and full montage (**b**) of a single GMM growth channel containing wild type (WT) cells carrying the circadian reporter *pkaiBC:eYFP-fsLVA*. The mother cell is outlined in white. Images were taken 45 min apart, with every second time point shown. Auto-fluorescence (red) and eYFP (reporter fluorescence, green) images are superimposed for display. **c, d** Single-cell length (**c**) and fluorescence (**d**) readouts from a representative medium light movie (n = 64 mother cell lineages). Representative lineage high-lighted by dark blue line. Dark blue shade indicates the 12 h dark entrainment pulse; lighter blue shades indicate subjective nights. Time point 0 corresponds to dawn. **e** Period distribution across lineages (measured as the period of the autocorrelation function) under three LL light conditions: low, medium, and high LL with photon fluxes of 10, 20, and 40 μmol m$^{-2}$ s$^{-1}$, respectively. Inverse relationship between light intensity and clock period is indicative of Aschoff's rule. Supplementary Fig. 3

provides fluorescence data for low (n = 284) and high LL (n = 108). Dashed vertical lines indicate mean periods for each light condition (mean period p ± standard deviation: p$_{low LL}$ = 25.5 ± 0.7 h, p$_{medium LL}$ = 24.6 ± 0.6 h, p$_{high LL}$ = 23.6 ± 0.6 h). **f** Phase diffusion time (left) and autocorrelation time (right) distributions across individual lineages for the three LL conditions. **g** Trajectory of clock expression noise (C.V.$^2$) as a function of reporter fluorescence level reveals a 'noise loop'. ZT$_0$ = zeitgeber time 0 (subjective dawn); period 1 corresponds to the 1$^{st}$ period after entrainment. Data shown for medium LL – see Supplementary Fig. 6 for low LL and high LL. **h** The clock is synchronised between and within lineages, as repre-sented by the high synchronisation index (top) and small phase difference error (bottom) between daughter cells irrespective of the phase of the mother cell at division. Phase was normalised to the range 0 to 1. Error bars represent the standard error of the mean.

previous report[36]. We note that we did not detrend our data, which is often required for movies taken on agarose pads due to the increase in bleed-through of fluorescence signal between neighbouring cells that occurs as a colony grows[32]. Even agarose pads micropatterned with channels do not keep a single line of cells after a few days[34]. Thanks to the simple geometry of the microfluidics setup allowing us to retain and image the daughter cells for several generations, we can also characterise the clock behaviour within cell lineages with high preci-sion. Previous work illustrated that cell division does not affect the clock timing and its inheritance between daughter cells[16]. Here, we further demonstrate precise clock phase inheritance at the point of cell division and show that phase differences between daughter cells are independent of the mother's phase at the time of cell division (Fig. 1h bottom panel).

## Known clock regulators are not required for clock robustness in the free-running clock

In order to understand the mechanisms contributing to clock preci-sion in vivo, we targeted four major regulators of the cyanobacterial clock previously identified in bulk studies as affecting the clock rhythm: *ldpA*[24,46], *pex*[25,47], *prkE*[26], and *lalA*[28] (Fig. 2a). *ldpA* and *pex* have been previously identified as responsible for synchronising the clock with the external light environment, although no specific mechanisms were elucidated[24,46,48]. LdpA is an iron-sulphur centre-binding protein thought to fine-tune the period of the clock in response to changes in photosynthetic electron transfer, which reflect changes in light intensity[24]. Pex was shown to affect the clock speed through negatively regulating *kaiA* expression[49]. *ldpA* and *pex* knockout (KO) mutants are reported to have a 1 h shortening of the period when compared to

**Table 1 | *S. elongatus* strains used in this study**

| Strain name | NSII integration | Antibiotic resistance | Origin/ Reference |
|---|---|---|---|
| WT | *pkaiBC:eYFP-fsLVA* | Kan | Locke lab |
| WT-Ab | *pkaiBC:eYFP-fsLVA* | Kan, Gent | This study |
| *ΔlalA* | *pkaiBC:eYFP-fsLVA* | Kan, Gent | This study/[28] |
| *ΔldpA* | *pkaiBC:eYFP-fsLVA* | Kan, Gent | This study/[24] |
| *Δpex* | *pkaiBC:eYFP-fsLVA* | Kan, Gent | This study/[47] |
| *ΔprkE* | *pkaiBC:eYFP-fsLVA* | Kan, Gent | This study/[26] |
| LP48 (KaiC-A251V) | *pkaiBC:eYFP-fsLVA* | Kan, Sp, St | This study/[51] |
| KaiC-R215C | *pkaiBC:eYFP-fsLVA* | Kan, Gent | This study/[17] |
| KaiC-T495A | *pkaiBC:eYFP-fsLVA* | Kan, Gent | This study/[17] |
| SP16 (KaiC-R393C) | *pkaiBC:eYFP-fsLVA* | Kan, Sp, St | This study/[51] |

The strains were constructed from the WT SynPCC 7942 background with the use of homologous recombination and antibiotic resistance selection. NSII stands for neutral site II. Kan stands for kanamycin antibiotic resistance (5 μg/ml), Gent for gentamicin (2 μg/ml), Sp for spectinomycin (2 μg/ml), St for streptomycin (2 μg/ml). All the strains were generated in the Locke group; references to the original work describing the phenotypes are given.

WT[24,47]. The involvement of *prkE* in light sensing was proposed following observations of erratic phase resetting in *prkE* KO mutants[26], but the mechanism remains elusive. Finally, we investigated one output clock effector, *lalA*, previously shown to influence the transcriptional feedback loop involving the *kaiBC* operon[28].

We generated KO strains of these individual genes and measured their free-running clock dynamics in individual cells using the GMM. We observed that removing each of those four clock regulatory genes resulted in changes in the clock free-running period (Supplementary Fig. 7). In particular, our single-cell findings of period shortening in *ΔldpA* and *Δpex* agree with previous observations in bulk[25,46]. However, despite the differences in reporter expression levels, the deviations from the WT free running period were minor. Raw trace inspection (Fig. 2b, see Supplementary Fig. 7 for period distributions) and autocorrelation analysis (Fig. 2c) revealed no major differences in the rhythm stability between WT clocks and the KO strains. We also observed long autocorrelation and phase diffusion times in most of the clocks of the KO strains, comparable to the WT's (Supplementary Fig. 8). We saw a slight decrease in autocorrelation time in the *ΔprkE* mutant, which might be indicative of the involvement of the gene in light sensing. This KO has been shown to impair the cells' ability to predictably adjust clock phase in response to light resetting conditions[26]. Noise loop shapes and noise magnitudes were largely conserved in all the KO mutants, with only a slightly elevated C.V.² in *Δpex* (Supplementary Fig. 9). Finally, the deletion of regulators affected neither the inter-lineage clock synchronicity nor phase inheritance at division (Fig. 2d). Overall, under constant light, the clock in vivo functioned well in the absence of these regulators, just like in vitro.

**A simple model of the KaiC phosphorylation loop explains how clock noise is affected by perturbations to the core clock network**

Next, we investigated how mutations in the core clock network influenced clock robustness. Multiple point mutations in the core clock gene, *kaiC*, have been observed to affect the clock rhythm[17,50–52]. We selected four mutants for further analysis, previously reported in bulk studies to display arrhythmia (KaiC-T495A), damped short period oscillations (KaiC-R215C), short period of 16 h (KaiC-R393C, henceforth SP16), or long period of 48 h (KaiC-A251V, henceforth LP48). We constructed these mutants in the background of our reporter strain and then observed their rhythms in the GMM under LL conditions. We

ensured that the phenotypes observed were driven by the perturbations to the clock and not the accompanying insertion of antibiotic resistance genes (Supplementary Fig. 10). The mean clock reporter trace in the SP16 and LP48 backgrounds had an accelerated and a slowed-down rhythm, respectively, as previously reported[51] (Fig. 3a, b). Our single-cell analysis revealed that these period changes are also associated with increased noise, as seen from both the raw data (Supplementary Fig. 11) and the autocorrelation analysis (Fig. 3b top). Interestingly, when normalised by period duration, the autocorrelation functions of the WT, the faster (SP16) and slower clocks (LP48) matched suggesting that noise levels are set over one period (Fig. 3b bottom, see Supplementary Fig. 12 for unscaled autocorrelation).

Single-cell investigation of the clock rhythmicity in KaiC-R215C and KaiC-T495A backgrounds revealed the existence of noisy individual oscillators, whose presence is masked by damped oscillations[17,50] in the mean trace (Fig. 3a, Supplementary Figs. 11,13). These noisy oscillations can also be observed in the plots of autocorrelation (Fig. 3b). In all the KaiC mutants we observed, the noisy oscillations also resulted in shorter phase diffusion times and autocorrelation times (Supplementary Fig. 14) when compared to the WT.

To understand the effects of genetic perturbations on KaiC on the noise properties of the clock, we adapted a stochastic model of the clock[45], which is based on deterministic models that generate a limit-cycle oscillation[45,53,54] (Fig. 3c). We parameterised the stochastic model using our single-cell data of mean period and C.V. in medium light conditions via approximate Bayesian computation (Supplementary Fig. 15, Methods). The model simulates the ordered transitions in the phosphorylation state of KaiC that occur across a circadian cycle, which are mediated by KaiA and KaiB. We note a phase difference between the transcriptional reporter in the experiments and the KaiC-P fraction in the model (Fig. 3a and c) due to gene expression downstream of KaiC that is not included in the model[53,55,56]. To provide a quantitative comparison between model and experiment, we fitted the mean and C.V. in the timing of phosphorylation peaks in the model with the mean and C.V. in the timing of troughs of reporter expressions that occur during the day (see Methods for details). This provided a good fit with the WT experiments under medium light conditions (Fig. 3d).

We then simulated changes in the clock period by perturbing several rates of the fitted WT model, motivated by the fact that core clock mutations can affect the KaiA binding domain in KaiC[17] or alter ATPase activity[51]. The model predicted that increasing the KaiA-KaiC unbinding and the KaiC phosphorylation rates causes a decrease in the clock period. Conversely, increasing the KaiA-KaiC binding and KaiC dephosphorylation rates increases the clock period. Clock perturbations that significantly accelerated or slowed the periods relative to WT were associated with higher noise levels (Fig. 3d, Supplementary Fig. 16). All parameter variations displayed a noise minimum around the fitted WT period (Supplementary Fig. 16). These predictions were in agreement with elevated noise levels we observed under genetic perturbations of the clock period (Fig. 3d). Taken together, our results show that WT clock robustness under constant conditions can be understood solely from considering the low-noise limit cycle oscillations generated through the KaiC phosphorylation loop.

**The stochastic model predicts noise buffering by the clock under increasingly complex environmental cycles**

Theoretical work has predicted that limit cycle oscillators can buffer input noise[8,54,57] motivating us to next examine clock robustness in individual cells under noisy light/dark (LD) cycles. These are also more representative of natural conditions, given that the clock has evolved under noisy periodic environments. We first studied clock robustness under the simplest entraining condition, 12 h: 12 h square LD cycles (Fig. 4a, Supplementary Figs. 17, 18 and Supplementary Movie 2).

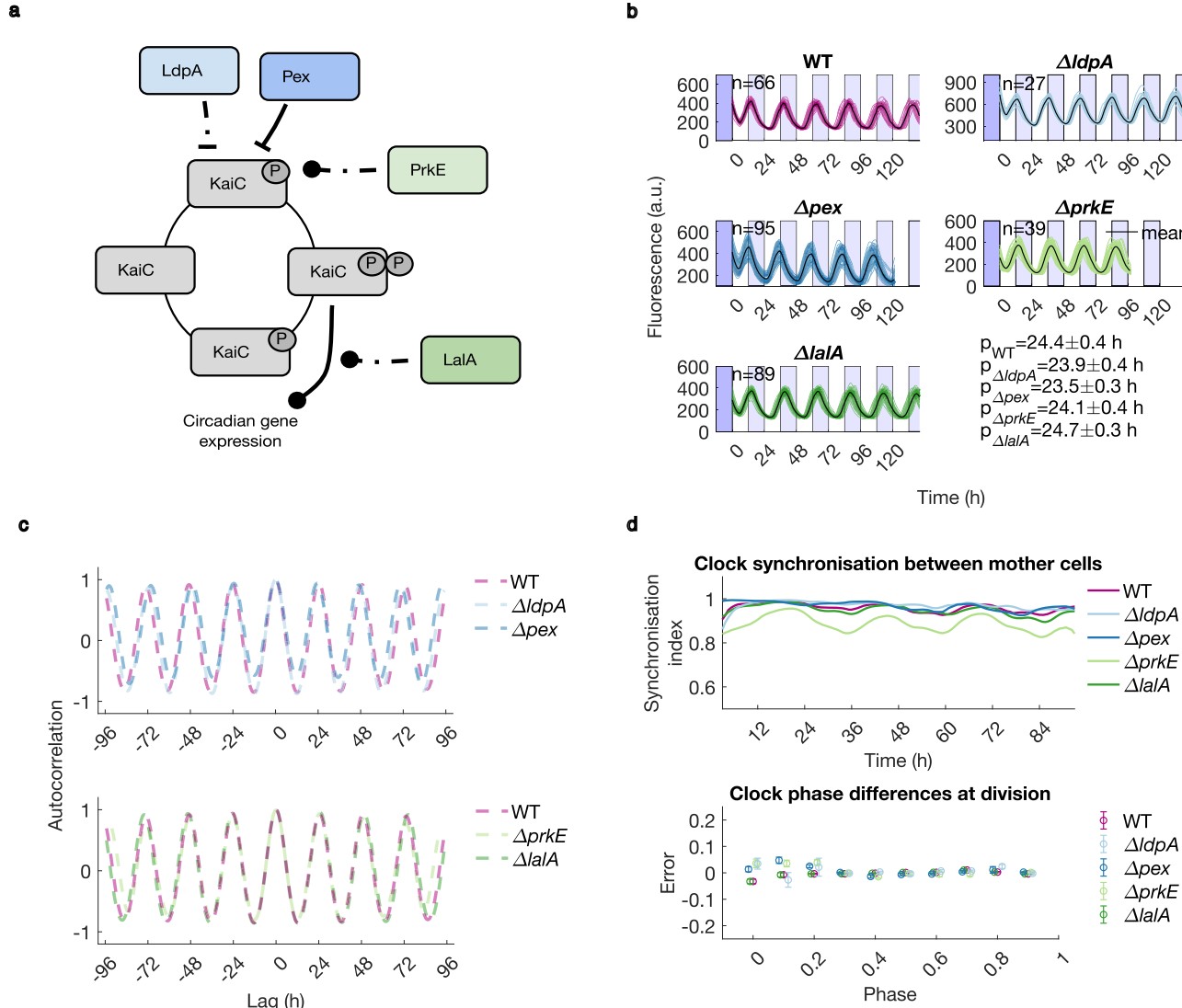

**Fig. 2 | Clock regulators are not required for free-running rhythm robustness in constant light. a** A schematic diagram of the clock regulatory network assessed in this study. Arrowheads represent regulation types: negative (-|) and undetermined (-o), all interactions are indirect. **b** Reporter fluorescence dynamics in individual cell lineages. n indicates the number of mother cell lineages analysed, and p indicates the mean period of the oscillation ± standard deviation (estimated from auto-correlation functions). **c** Autocorrelation analysis reveals the stability of the clock rhythm in the absence of individual clock regulators under LL (see Methods for details). The same WT dataset is plotted in both graphs. **d** The synchronicity between mother cells remained high in the KO mutants. Similarly to the WT, the phase of the clock (normalised to between 0 and 1, shifted along the x-axis for visualisation purposes) was accurately inherited across cell divisions. Error bars represent the standard error of the mean. Data and n (number of mother cell lineages) for **c**, **d** are as described in Fig. 2b. See Fig. 1h and Methods for quantification details. Cells were imaged under medium LL (20 μmol m⁻² s⁻¹).

Robust 24 h oscillations were observed under two different light intensities.

Next, we attempted to simulate clock dynamics under LD cycles using the stochastic model. The model demonstrates entrainment of clock traces to the environmental period when phosphorylation rates are assumed to vary with light intensity. Our simulations successfully capture phase variation of clock lineages across multiple days and its corresponding decrease with peak light intensity (Fig. 4b) after adjusting the dependence of phosphorylation rates on light intensity via an effective Michaelis-Menten function informed by the experimental data (Methods). In agreement with the model, our data display the signatures of phase diffusion, a linear increase of phase variance over successive days, in free-running conditions, and signatures of entrainment, a plateau of phase variance, under LD conditions.

In the wild, light patterns are heavily affected by unpredictable daily weather conditions. We used the model to predict the clock behaviour under both artificial environmental noise and realistic meteorological conditions. We first simulated a simple noisy perturbation in the timing of the light-dark transition (LD switch) in square-wave LD cycles. In 'noisy day start' conditions we delayed the onset of light by up to 3 h, and in 'noisy day end' conditions we delayed the transition from light to dark by up to 3 h and accelerated it by up to 1 h (Fig. 5, Supplementary Fig. 19). Such perturbations were generated arbitrarily using a random number generator function sampling from a uniform distribution.

Using the model, we predicted that the mean clock trough time would shift only by a fraction of the environmental perturbation and that this response varies little from cell to cell, buffering the noise it is exposed to (Fig. 5b). Our experimental data confirms the prediction of our model, which was parameterised only under non-noisy light conditions (Fig. 4a, Methods). The agreement is observed both for the mean trough times and the trough time variability within the

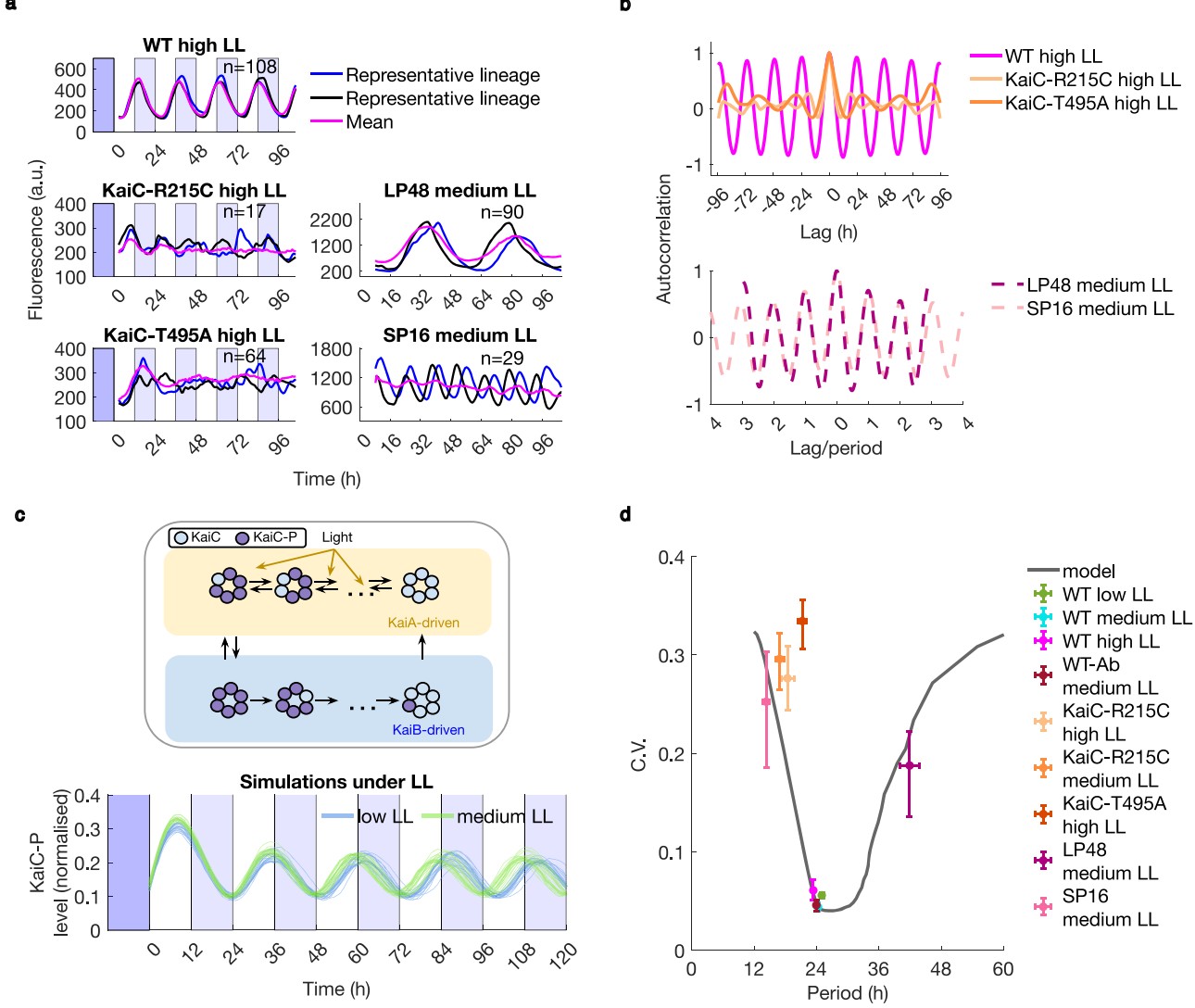

**Fig. 3 | Robustness of oscillations to genetic perturbations in the core clock.**
**a** Clock dynamics in individual lineages in mutant backgrounds reveal residual oscillations which are masked by bulk data averaging. WT, KaiC-R215C and KaiC-T495A lines were imaged under high light (40 µmol m⁻² s⁻¹) whilst SP16 and LP48 were imaged under medium light (20 µmol m⁻² s⁻¹). WT high LL is the same dataset as in Fig. 1. n is the number of mother cell lineages. Fluorescence values presented for medium light conditions were obtained using different imaging settings (see Supplementary Table 1 for details). Two representative lineages are shown for each strain. All individual cell lineages for all strains displayed are shown in Supplementary Fig. 11. Time point 0 corresponds to dawn. **b** Auto-correlation of individual lineages reveals residual oscillations of non-24 h periods in different mutants, confirming and extending previously published reports of clock behaviour in bulk in corresponding genetic backgrounds. **c** Top: Model cartoon based on Chew et al.[45]. In our updated version of the model, light intensity during the day biases KaiC phosphorylation (KaiC-P is the phosphorylated form). **Bottom**: Representative lineages of KaiC phosphorylation level

simulated using the stochastic model (Methods) under two different light intensities, following Aschoff's rule. **d** Perturbations to dephosphorylation rate in our model (black line represents the best fit) qualitatively describe the increase in clock noise (C.V.) associated with the period changes in the trough time distributions of mutants chosen in this study (points). All WT data under different LL conditions collapse to the local minimum on the graph. Similar noise minima were observed for other model perturbations (Supplementary Fig. 16). Error bars, centred on the mean, represent 95% confidence intervals obtained by bootstrapping of trough time distributions. WT Low, Medium and High LL data correspond to the data introduced in Fig. 1, WT-Ab data is shown in Supplementary Fig. 10, KaiC-R215C medium LL data is shown in Supplementary Fig. 13 to demonstrate that the noise properties induced by the changes in clock genetics are persistent under different light levels, and remaining data are as in Fig. 3b. n, number of mother cell lineages analysed, is as listed in the figures where the corresponding data is first presented.

population of oscillators (Fig. 5a, b, Supplementary Fig. 20 right column shows C.V. of the mean trough time distributions). Moreover, we observed that the noise levels in the clock were relatively stable across the range of perturbations, although the noise levels produced by the model for noisy day end conditions were lower than in the data (Supplementary Fig. 20 left column). Together with the buffering of the shifts in the mean trough time, this noise robustness illustrates the resilience of the clock to perturbations in the timing and amounts of light.

Next, we simulated the behaviour of the clock under environmental cycles that represent real environmental conditions, using meteorological data from Abanico, Varadero - a Caribbean coral reef in Colombia[58]. We chose two sets of five and four consecutive measurement days (January 7th–11th 2017, Caribbean 1, and Dec 1st–4th 2017, Caribbean 2, respectively) as these were the closest in their cumulative light intensity to the square LD cycles in our experiments. These meteorological light profiles include high-frequency noise and a large day-to-day variation in light amount, as illustrated in Fig. 6a and

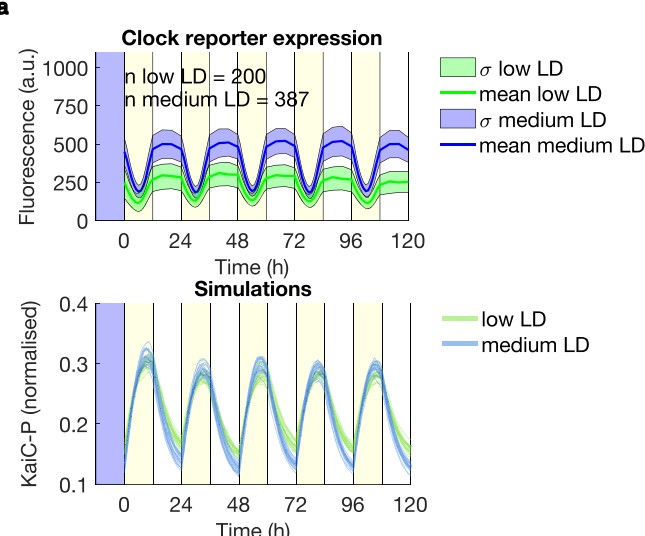

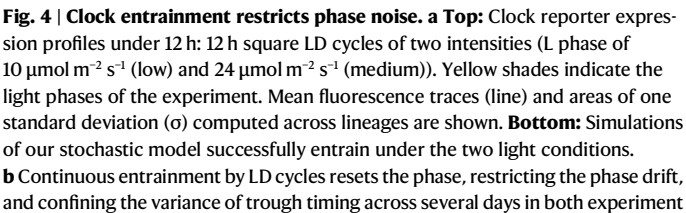

**Fig. 4 | Clock entrainment restricts phase noise. a Top:** Clock reporter expression profiles under 12 h: 12 h square LD cycles of two intensities (L phase of 10 μmol m$^{-2}$ s$^{-1}$ (low) and 24 μmol m$^{-2}$ s$^{-1}$ (medium)). Yellow shades indicate the light phases of the experiment. Mean fluorescence traces (line) and areas of one standard deviation (σ) computed across lineages are shown. **Bottom:** Simulations of our stochastic model successfully entrain under the two light conditions. **b** Continuous entrainment by LD cycles resets the phase, restricting the phase drift, and confining the variance of trough timing across several days in both experiment and simulation. Under LL, however, variance increases linearly with time (shown as a function of trough-to-trough distance). Low LL and LD conditions are shifted along the *x*-axis for visualisation purposes. Error bars represent 95% confidence intervals obtained by bootstrapping of trough time distributions. LL data correspond to the experiments presented in Fig. 1 (low and medium LL) and Fig. 3 (medium LL). n, number of mother cell lineages, are as listed in these figures. For more LL data displaying linear increase of phase variance with time, see Supplementary Fig. 18.

Supplementary Fig. 21. Capturing both the response to temporal variations and the absolute irradiance at the site represented a challenge to our model parameterised only on data from non-noisy light conditions. The model accurately predicted both the trough timing and its variability under both sets of conditions (Fig. 6b). The noise (C.V.) in trough times was 9–12%, which is similar to the 8–10% we measured under non-noisy square wave LD cycles. We found that the clock filtered out the high-frequency variation in both the experiment and model (Fig. 6c, d), whilst changes in the trough time indicated that the clock responded to the day-to-day variation in light intensity.

Finally, we tested our predictions of the clock filtering fast environmental fluctuations by applying 12 h: 12 h square LD cycles with high-frequency (every 10 min) light variations of up to 25% around the mean light level (24 μmol m$^{-2}$ s$^{-1}$). We found that the clock filtered out these high-frequency rhythms, responding similarly to non-noisy LD cycles with a steady trough time distribution, both in the simulations and the experiments (Supplementary Fig. 22). This also fits with the theoretical predictions for noise filtering by limit cycle oscillators[8,57]. The continuous modulation of the phosphorylation reactions in our model represents a mechanism by which the clock filters noise (Methods). These reactions are slower than the noise frequency, implementing a low-pass filter that confers robustness to high-frequency environmental noise.

## Discussion

In this paper, we have developed a new framework to assess the robustness of circadian rhythms in individual cells. We adapted a bacterial microfluidics setup to develop the Green Mother Machine (GMM), which allows cultivation and imaging of single-cell cyanobacteria under a stable micro-environment. Using this method, we quantified the robustness of the cyanobacterial circadian clock to a series of genetic and environmental perturbations. We showed that the clock is robust to removal of four clock regulators under constant light conditions, and revealed the true dynamics of mutations to the core clock component KaiC that were averaged out in previous bulk studies. Combining experiment and modelling, we found that a simple model

of the interactions in the core phosphorylation loop network of KaiA, B, and C can predict how the clock buffers noise in the timing of the start of the day and night, as well as fast fluctuations under natural environments. Our work reveals how the balance between robustness and plasticity is set in the cyanobacterial clock.

We first examined clock robustness under a series of constant light conditions (Fig. 1). We found the clock to show either similar[45] or smaller[16] phase diffusion (Fig. 1f), and lower noise amplitude[36] (Fig. 1g) than previously reported, potentially due to the improved stability and homogeneity of growth conditions in the GMM. We also observed that the clock ran faster with higher light levels (Fig. 1e), in line with Aschoff's rule for diurnal organisms[41]. While we did not vary steady-state light levels within the course of individual experiments, our data imply that individual clocks adjust their speed in response to changes in light intensity. This is further supported by the shifts in clock phase that we observe under our natural light/dark conditions, where in general the phase is advanced or delayed in days with higher and lower light, respectively (Fig. 5).

We then looked at how the WT clock robustness persists under genetic perturbations (Fig. 2). The free-running rhythm appeared to be robust to the deletion of all four clock regulators we investigated. While this is not too surprising, given the oscillatory behaviour of the KaiABC complex in vitro[18], the prevalence of the wider clock regulatory network in vivo suggests the importance of clock regulators in the wild. In future work it will be important to inspect the roles of such genes in noisy light/dark environments, particularly *ldpA* and *prkE*: past experiments in bulk suggest their importance in fine-tuning the clock response to light intensity variability[24] and phase resetting stimuli[26], respectively. Turning to the core clock network, we found mutations to KaiC influenced the noise both within and between individual lineages (Fig. 3). A simple stochastic model of the KaiC phosphorylation state loop was able to recapitulate the observed changes to the mutant clock dynamics, revealing an evolutionary optimum in the noise levels of the natural (ca. 24-h period) WT clocks under a range of light intensities (Fig. 3d). The noise minimum was robustly observed across model perturbations, which implies that WT

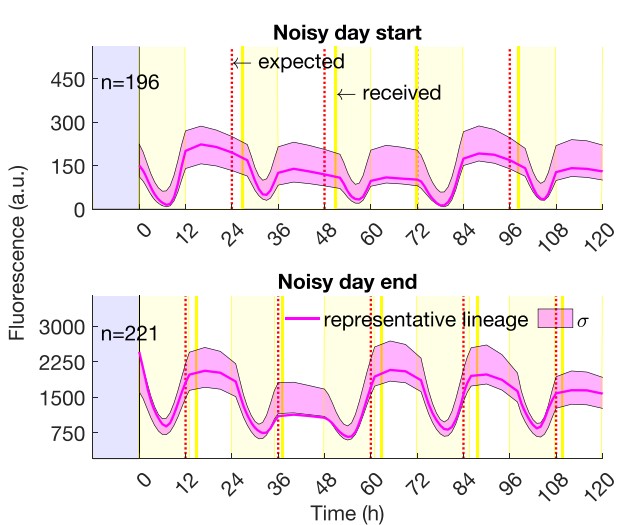

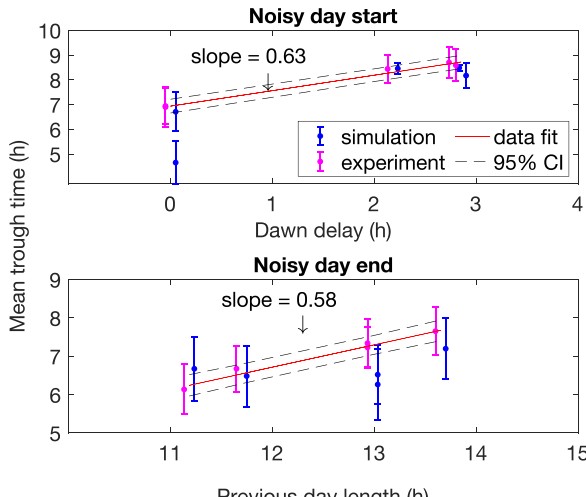

**Fig. 5 | The clock buffers noise in day start and day end timing, as predicted by the model. a** Clock reporter data from the noisy LD switch under square LD cycles. Yellow shades indicate the light phases of the experiment; the expected and the received timings of the switch are shown in dashed red and solid dark yellow lines respectively. For details on the timing of the switch between light and darkness and the complete fluorescence data see Supplementary Fig. 19. n indicates the number of mother cell lineages analysed; a representative lineage and the data distribution within one standard deviation from the mean (σ) are shown. Differences in observed reporter expression levels are driven by the differences in imaging setups (see Supplementary Table 1). **b** The model predicts, and the data confirm, that the clock responds to the perturbation with a partial shift in phase (slope <1), buffering the noise in the environment. Linear fit to data and the corresponding 95% confidence intervals (CI) are shown. Data and n, the number of mother cell lineages, are the same as in Fig. 5a. Data points representing the mean for simulation and experiment are slightly shifted along the *x*-axis for visualisation purposes. Error bars represent one standard deviation of the trough time distribution.

clocks function as low noise limit-cycle oscillators while long and short period mutants operate closer to Hopf bifurcation thresholds with oscillations sustained by larger noise levels[10]. Such elevated noise levels induce damping of oscillations at the population level. Indeed, bulk studies showed that both short and long period mutants, including the mutants we examined, displayed more strongly damped bulk oscillations than the WT[50,51]. Other core clock mutations, which have been reported only recently and we did not test, appear to cause period changes without significant oscillation damping at bulk level[52], although a quantitative characterisation of damping is still lacking. These mutations could involve mechanisms we did not model and would be interesting to investigate using our model or a modified version thereof. Taken together, our data suggest the noise properties of the clock can be understood simply in terms of the interactions between KaiA, B and C, without needing to consider the feedback from KaiABC on its own transcription (Figs. 3, 4). Indeed, a recent study showed that low period noise observed in the WT under constant conditions is reproduced by an inducible expression system, with no transcriptional feedback, when expressing the three Kai proteins at WT levels[45]. However, a previous study suggested that the transcriptional loop reduces clock noise[34]. Future work should clarify under which conditions the transcriptional feedback loop is required for rhythm robustness.

Applying noisy perturbations to the timing of environmental transitions, both at dawn and at dusk, we observed the clock adjusts its phase proportionally to the difference in time between the expected and actually observed light change (Fig. 5). Mean reporter trough times occur approximately 0.6 h later per 1 h delay in the transition timing (Fig. 5b). This is reminiscent of a previous study by Leypunskiy et al.[39], which found a similar relation between clock phase and day length when cells were stably entrained to a range of photoperiods. Their results indicate the clock is able to gradually adjust its phase to track midday. While midday tracking explains the ability of the clock to adjust to gradual seasonal changes in light profiles, our findings under noisy day start/end conditions suggest that the relation between phase delay and day length is a more fundamental clock feature that can be utilised to buffer against unpredictable one-off fluctuations.

Examining clock behaviour under conditions mimicking natural light profiles, we found, in both the model and experiments, that the clock filters high frequency noise (Fig. 6, Supplementary Fig. 22). In the model, noise buffering occurs due to the slow phosphorylation reactions acting as a low-pass filter. These slow rates fit with biochemical estimates that each KaiC molecule consumes less than 1 ATP molecule every hour during the phosphorylation cycle[51]. However, further work is required to reveal exactly which components of the phosphorylation cycle act as the noise filtering step and what role clock regulators can play for noise buffering in vivo. It will also be important to examine in more detail the relationship between light intensity and phosphorylation rates, for example through linking these with ATP/ADP dynamics that were not included in our model[21].

The noise properties we examined in this study may also apply to higher organisms and more complex clocks. For example, clock behaviour under different day lengths has been studied in detail in plants, but not in individual cells[59]. The plant molecular clock network has a multiple feedback loop structure, as do the clocks of other higher eukaryotes. These feedback loops respond differently to changes in day length, with the phase of some loops tracking dawn and the phase of other loops tracking dusk[60]. To what extent these loops have evolved to buffer cell-to-cell noise at different times of the day is an open question. Future work in the cyanobacterial clock should also be concerned with elucidating the functional consequences of the noise buffering. A previous study suggests that the clock orchestrates glycogen metabolism to build up cellular endurance during periods of energy limitations at night[40]. Future studies could look into the relationship between the clock state and the energy storage under natural environments.

Our work shows the power of single-cell analysis in revealing robustness and plasticity properties that are often missed in bulk. Although we focused on light perturbations in the GMM, in future it will be possible to examine clock rhythms under nutritional and

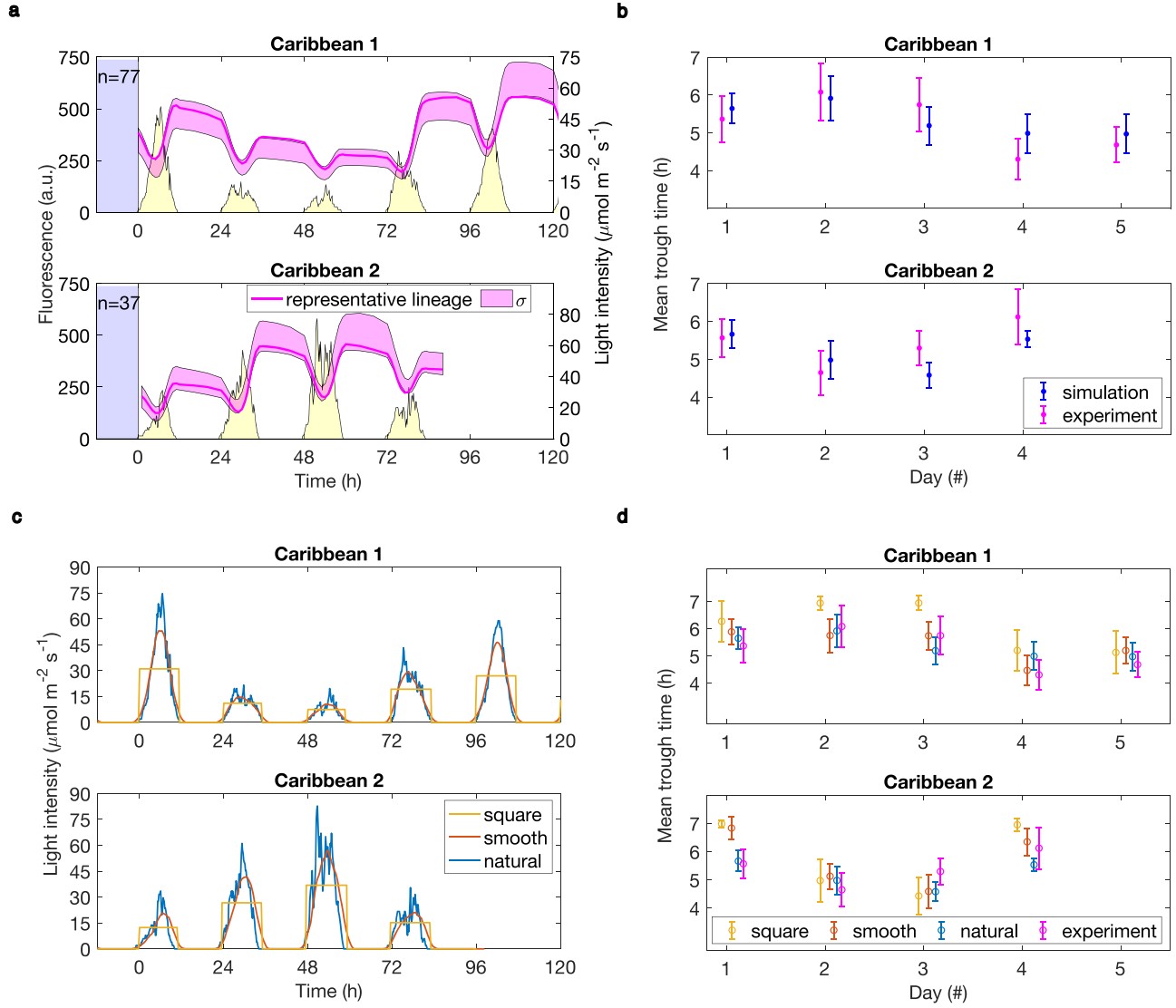

**Fig. 6 | The model predicts clock timing and high-frequency noise filtering under meteorological conditions. a** Clock dynamics (magenta) under meteorological light conditions matching absolute irradiance and temporal profiles measured at the Abanico coral reef, Colombia (yellow). n indicates the number of mother cell lineages; a representative lineage and the data distribution within one standard deviation from the mean ($\sigma$) are shown. For details on light intensities and raw data see Supplementary Fig. 21. **b** Model predictions and experimental validation of the trough time dynamics in response to meteorological conditions. Data points for simulation and experiment are shifted along the x-axis for visualisation purposes. Error bars, centred on the mean, represent one standard deviation of the trough time distribution. Overall dynamics indicate that the clock slows down in response to lower light days (later trough times) and speeds up under higher light (earlier trough times). **c** Simulations of the Caribbean light conditions ('natural') and their denoised versions, either smoothed via moving averages or as a square wave of equal integral daily intensities. Total daily light intensity is equal under all three conditions. **d** Mean trough time of the clock reporter under noisy and non-noisy conditions in the simulations is comparable to the experiments with noisy light inputs. Error bars represent one standard deviation of the trough time distribution. Data and n, number of mother cell lineages, are as shown in Fig. 6a.

chemical perturbations. Furthermore, the GMM will be a general tool for examining cyanobacterial gene expression, allowing the characterisation of synthetic biology parts and other key biological processes, as has been the case for the Mother Machine in *E.coli*[35,61,62]. Our measurements of the low amplitude noise (Fig. 1g) and little phase discrepancies between and within daughter cells (Fig. 1h) confirm the exceptional robustness of the cyanobacterial clock in free-running conditions. Current synthetic oscillators, such as the repressilator[63] (and its most advanced form[61]), dual feedback oscillators[64] (and their versions optimised using genetic screens[65]), or the *kaiABC* system transplanted into *E.coli*[66] do not achieve such high stability. A multifaceted systems approach, such as the one we followed here, has the power to reveal further insights into the principles of noise control underlying cell-to-cell synchronicity, allowing us to advance our

understanding of endogenous oscillators and the design of synthetic circuits.

## Methods

### Strains and growth conditions

*S. elongatus* strains used in the study (Table 1) were generated upon transformation and subsequent homologous recombination. Chemically competent *E.coli* DH5α and HST08 cells were used as hosts for molecular cloning via common molecular biology techniques. An established transformation protocol[67] was followed. Chromosomal integration of transcriptional reporters and point mutations were confirmed upon PCR and Sanger sequencing.

The reporter strains had degradation-tagged fluorescent proteins expressed from a clock promoter *pkaiBC*, with constructs inserted in

neutral site II (NS II)[68]. The reporter used in this study (*pkaiBC:eYFP-fsLVA*) has a frameshift mutation that results in intermediate degradation rates of the clock reporter (a half-life of 4.1 h, (CI, 95% confidence intervals: 3.9, 4.3 h))[44], allowing robust imaging of clock dynamics. Strains are available upon request.

Gene deletion was achieved by insertion of an antibiotic resistance cassette by homologous recombination to disrupt the gene of interest. Plasmids pTS82, pTS83, pTS86 and pTS87 were constructed to carry out this method (Supplementary Data 1).

Single point mutations of the *kaiC* gene were obtained by first constructing base plasmids containing the antibiotic resistance cassette immediately downstream of the *kaiC* gene. The *kaiC* gene in the base plasmid was then mutated by site-directed mutagenesis using the primers described in Supplementary Data 1 to obtain plasmids pTS63, pTS64, pTS93 and pTS94. Transformation of *S. elongatus* with these plasmids produced a single point mutation of the *kaiC* gene in the endogenous locus. Mutation of the *kaiC* gene was confirmed by Sanger sequencing of an amplicon of the gene from the genomic DNA of the strain. Transformation with the base plasmids allowed for control of the presence of the antibiotic resistance cassette immediately downstream of the non-mutated *kaiC* gene (WT-Ab strain).

The strains were grown in BG-11 M medium under 25 µmol m$^{-2}$ s$^{-1}$ cool white light[69]. Percival chambers were used to maintain the growth environment at 30 °C with 30% relative humidity. Cells were cultured from the glycerol stocks until the late exponential or early stationary phase (OD$_{750}$ 0.8-1.3), then re-diluted to OD$_{750}$ 0.1 and grown until the mid-exponential phase (OD$_{750}$ 0.4-0.6) during entrainment. Strains were entrained by exposure to one 12 h:12 h LD cycle in liquid culture. Then, cells were loaded into the GMM and were exposed to additional 12 h of darkness in the microscope before data acquisition began. Clock period mutants were not entrained and were only imaged in constant light conditions.

## Microfluidics

The design and fabrication of the microfluidic chip used in this study was described in detail in Sachs et al.[42]. Briefly, the chips were fabricated from an epoxy master by casting a 10:1 base to a curing agent mixture of Sylgard 184 polydimethylsiloxane (PDMS) (Dow Corning, USA) onto the mould and cured for 2.5 h at 65 °C. To remove uncured PDMS, the chips were first washed in 100% pentane (Sigma-Aldrich, USA) for 90 min, followed by two washes in 100% acetone (Sigma-Aldrich, USA) for 90 min, and then dried overnight[70]. On the day of the experiment, the chips were bonded to a glass bottom dish (HBSt −5040, Wilco Wells, Netherlands) using plasma treatment (Femto Plasma System, Diener, Germany). The bonding was strengthened by baking the chips at 65 °C for 10 min. Prior to cell loading, the chips were passivated with 0.1 mg/ml PLL-g-PEG (SuSoS, Germany) for 20 min at 37 °C.

Cells grown in BG−11 M media (10 ml) were concentrated from bulk cultures by centrifugation (3000 g for 8 min) and injected into the GMM inlets. To force the cells into the growth channels, the chips were spun at 3000 rpm for 5 min using a spin coater (Polos Spin150i, SPS, Netherlands). The protocol did not result in every channel of the chip being filled, but allowed sufficient loading without damage to the cells. Media was supplemented with 0.1 mg/ml BSA (Sigma Aldrich, USA) and supplied through PTFE tubing (Darwin Microfluidics, France) at a flow rate of 0.12 ml/h by a syringe pump (Fusion 100, Chemyx, USA). We found that this choice of tubing material was critical for healthy growth and survival of cyanobacterial cells in multi-day microfluidic experiments.

## Microscopy and image analysis

Images were acquired using an inverted light microscope (Nikon Ti-eclipse, Nikon, UK) equipped with phase contrast, epifluorescence imaging modules, and the Nikon Perfect Focus System (PFS). All data

were obtained with a Nikon 100x Plan Apo (NA 1.4) objective in combination with Leica Microsystems immersion oil type F (Leica, Germany) using a Photometrics Prime sCMOS camera and a CoolSNAP HQ2 camera (Photometrics, USA). See Supplementary Table 1 for the imaging conditions for all the datasets presented. To reduce phototoxicity, the maximal imaging frequency was set to every 45 min (LL) or 60 min (LD) under light and every 240 min under darkness.

Excitation light was provided by a Lumencore Solar II light engine (Lumencore, USA) and Chroma filters (Chroma, USA) #41027 and #49003 were used for the 'red' (auto-fluorescent) and the YFP channels, respectively. A circular cool-white light LED array (Cairn Research, UK) was used to provide light for photoautotrophic growth. Its light spectrum, measured using a LI−180 spectrometer (LI-Cor, USA), is shown in Supplementary Fig. 23. An incubation chamber (Solent Scientific, UK) was used to maintain a 30 °C temperature. The microscopy setup was controlled with MetaMorph Software (Molecular Devices, USA).

Image processing was conducted in MATLAB (MathWorks, USA) using code (available upon request) adapted from Schwall et al.[43]. The approximate background fluorescence value (i.e. mean pixel intensity) of an empty channel, 200 for images obtained with the CoolSNAP camera and 300 for images obtained with the Prime sCMOS camera, was subtracted prior to reporter dynamics analysis.

## Data analysis

Hilbert transform (HT) was applied to LL data (Supplementary Fig. 4) to quantify the instantaneous clock phase $\varphi_j$ (running between 0 and 1) in the *j*-th lineage and compute the corresponding phase diffusion coefficient $D_j$, via linear regression of

$$Var_t\left(\varphi_j(t+\Delta t) - \varphi_j(t)\right) = D_j \Delta t \qquad (2)$$

and phase diffusion time

$$T_j = 2/D_j \qquad (3)$$

in units of days (Fig. 1f). Lineage periods $T_j$ and autocorrelation times $\tau_j$ were computed from a fit of

$$A_j \cos(2\pi t/T_j)e^{-t/\tau_j} \qquad (4)$$

to the autocorrelation function damping as described in the Supplement (Supplementary Fig. 4). The synchronisation index (Figs. 1h and 2d) was computed as in Teng et al.[34]:

$$S(t) = \frac{1}{M(t)}\sum_{j=1}^{M(t)} e^{i2\pi\varphi_j(t)} \qquad (5)$$

where $M(t)$ is the number of lineages existing at time point $t$.

Reporter peaks occur during the night and cannot be timed accurately under LD due to the transcriptional shutdown in the cell[71]. A consequent increase in photobleaching sensitivity motivates the reduction of imaging frequency in the dark[31] and further decreases the reliability of peak detection. Hence, reporter troughs were analysed instead. Trough timing was analysed using Mathematica's peak finder function by multiplying the fluorescence traces by −1 after smoothing using a Gaussian blur (standard deviation of 2.25 h). High frequency troughs were then filtered by iteratively optimising a threshold for minimum trough distance corresponding to half the mean trough-to-trough distance until convergence.

## Mathematical modelling

Our stochastic model derives from the ordinary differential equation (ODE) model of phosphorylation kinetics of KaiC hexamers introduced

before[45]. We converted the ODEs into the following set of mass action reactions describing KaiA binding to KaiC, KaiA and KaiB-dependent phosphorylation, as well as spontaneous and KaiB-dependent dephosphorylation. KaiC is a hexamer, which exists in one of N phosphorylation states ($i=1$: unphosphorylated, … $i=N$ fully phosphorylated). Free KaiB is not a variable in the model.

1. KaiA un-/binding

$$A + C_i \underset{k_{off}}{\overset{k_{on}}{\rightleftharpoons}} AC_i,\ i = 1, .., N \tag{6}$$

$$6A + BC_i \xrightarrow{k_{ABC}} ABC_i,\ i = 1, .., N \tag{7}$$

2. KaiA-dependent phosphorylation

$$AC_i \xrightarrow{k_{phos}} AC_{i+1},\ i = 1, .., N - 2 \tag{8}$$

$$AC_{N-1} \xrightarrow{k_{phos}} BC_N + A \tag{9}$$

3. Spontaneous dephosphorylation

$$C_i \xrightarrow{k_{dephos}} C_{i-1},\ i = 2, .., N \tag{10}$$

4. KaiB-dependent dephosphorylation

$$ABC_2 \xrightarrow{k_{dephos}} C_1 + 6A \tag{11}$$

$$BC_2 \xrightarrow{k_{dephos}} C_1 \tag{12}$$

$$BC_i \xrightarrow{k_{dephos}} BC_{i-1},\ i = 3, .., N \tag{13}$$

$$ABC_i \xrightarrow{k_{dephos}} ABC_{i-1},\ i = 3, .., N \tag{14}$$

$$BC_N \xrightarrow{k_{dephos}} C_{N-1} \tag{15}$$

The reactions conserved the total amount of KaiA and KaiC via the following conservation relations:

$$[A_T] = [A] + \sum_{i=1}^{N} [AC]_i + 6\sum_{i=1}^{N} [ABC]_i \tag{16}$$

$$[C_T] = \sum_{i=1}^{N} ([AC]_i + [ABC]_i + [BC]_i + [C]_i) \tag{17}$$

The output of the model is the phosphorylation level defined by

$$p = \frac{1}{N-1} \sum_{i=1}^{N} (i-1)\pi_i \tag{18}$$

where

$$\pi_i = \frac{[AC]_i + [ABC]_i + [BC]_i + [C]_i}{[C_T]} \tag{19}$$

similar to other models[53,72].

The model was parametrised using ABC rejection sampling[73] fitting to a trough timing with CV = 0.044 in constant medium light intensity and rescaling all parameters to match the experimental mean period (trough-to-trough distance) of 24.25 h. $N = 6$, $[A_T] = [C_T] = 2400$ were fixed during sampling. Representative parameters from fitting used here are $k_{on} = 3.57 \times 10^{-4} h^{-1}$, $k_{off} = 2.14 \times 10^{-2} h^{-1}$, $k_{ABC} = 1.62 \times 10^{-4} h^{-1}$, $k_{phos} = 4.90 \times 10^{-1} h^{-1}$, and $k_{dephos} = 4.29 \times 10^{-1} h^{-1}$, while in low light LL conditions, we adjusted $k_{phos} = 4.80 \times 10^{-1} h^{-1}$. We fitted a light-dependent phosphorylation rate using the two experimental LD conditions shown in Fig. 4, which resulted in

$$k_{phos}(L) = k_m \frac{L}{K_L + L} + k_0 \tag{20}$$

with, $k_0 = 0.05 h^{-1}$, $k_m = 1.28 h^{-1}$, and $K_L = 0.8$, in scaled light units (which corresponds to ~12 µmol m$^{-2}$ s$^{-1}$). For comparison with experimental reporter expression in LD conditions, trough times in our clock reporter were compared to simulated peak times in phosphorylation levels after subtracting a constant offset of 2.5 h (Supplementary Fig. 22). The model was then used to predict clock timing under random light environments. The stochastic simulations were carried out using Gillespie's direct method in Catalyst.jl[74]. We assumed piecewise constant light intensities under Caribbean environments, as in the experimental light inputs, and propensities were updated at the corresponding time intervals.

## Reporting summary

Further information on research design is available in the Nature Portfolio Reporting Summary linked to this article.

## Data availability

Data required to regenerate the figures are available via: https://github.com/SashaEremina/clock_noise. Data has also been deposited at Figshare[75]: https://doi.org/10.6084/m9.figshare.28236155 and is publicly available as of the date of publication.

## Code availability

Scripts required to regenerate the figures are available via: https://github.com/SashaEremina/clock_noise. Extensible Julia code to run stochastic simulations with time-varying light inputs is available at https://github.com/pthomaslab/GillespieClockModel.jl. Code has also been deposited at Figshare[75]: https://doi.org/10.6084/m9.figshare.28236155 and is publicly available as of the date of publication.

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

## Acknowledgements
We would like to thank Dr Chao Ye for constructing the fluorescent reporter plasmid and aiding with construction of some of the strains designed in this study. Thank you to Mr Toby Livesey who took part in optimising the microfluidics setup. AE, CS, TS, and JL were supported by a fellowship from the Gatsby Foundation (GAT3272/GLC to JL). AE, PT, and JL were also supported by the Leverhulme Research grant (RPG-2022-328). PT was supported by UKRI through a Future Leaders Fellowship (MR/T018429/1). LW was funded by the Deutsche Forschungsgemeinschaft (DFG, German Research Foundation) – Project ID 458090666 / CRC1535/1. BM was supported by a Biological Sciences Research Council Grant (BB/V016628/1) and by a School of Life Sciences, University of Warwick start-up grant.

## Author contributions
A.E., J.C.W.L., B.M., and P.T. designed the study, analysed and interpreted the results, wrote the article. A.E., C.S., L.W., and D.K. developed the microfluidic setup. T.S. constructed the strains. A.E. conducted experiments. P.T. created the clock model. All authors provided input into the manuscript.

## Competing interests
The authors declare no competing interests.
