## [Transparent Peer Review file · Nature Communications]

Environmental and molecular noise buffering by the cyanobacterial clock in individual cells

Corresponding Author: Professor James Locke

Version 0:

Reviewer comments:

Reviewer #1

(Remarks to the Author)

There is much to like in the manuscript from Eremina et al. which uses a single cell analysis of circadian rhythms in bacteria to go beyond previous work which has touched on similar topics of stochasticity and response to external fluctuations. I see a major advance of this work to be the development of the "green mother machine" which required non-trivial technical innovation and allows for the first single cell measurements of the cyanobacterial clock under truly constant culture conditions.

The study uses this mother machine tool to address multiple interconnected questions. I have some major concerns about some of the conclusions as well as more minor suggestions.

Major concerns:

1. Many readers will assume that the title "Environmental and molecular noise buffering by the core phosphorylation loop..." refers to conclusions about cells that have removed the transcriptional feedback connections between the Kai proteins and their expression. In the lingo of the field, cells that only have a "PTO". But as far as I can tell all of the experiments are on cells with intact transcriptional feedback. This seems especially relevant since the Teng et al. paper argues that the transcriptional feedback plays a crucial role in stochasticity. The title should be changed and the relationship of the analysis to previous work should be clarified.

2. The observation that two selected period mutants cause an increase in stochasticity is quite interesting, The comparison with the model in Figure 3D is questionable, since it relies on two mutants previously reported to have period effects (and the "arrhythmic mutants" where period estimation is more difficult). The model treats all period effects as arising from changes in the rates associated with KaiA-KaiC interaction, but this seems implausible since residues 251 and 393 are quite far from KaiA interaction sites. Presumably there are other ways parameters could be changed in the model to adjust period that might produce different c.v. vs. period curves.

Perhaps another way to approach this would be to compare published (bulk) amplitudes for period mutants (see Ito-Miwa et al. for some dramatic examples)—do these agree with the Figure 3D curve?

More minor:

3. Something seems off with Fig 3B. It seems like the difference between the top and bottom panels is rescaling of the time axis by the oscillator period. Why do the relative amplitudes of the autocorrelation peaks seem to change?

4. Are the data in Fig 4B showing that phase coherence is improved at higher light at odds with data in the Teng et al. paper which seems to show increased stochasticity at higher light levels? Please comment

5. The experiments in Fig 5 are quite interesting. In many papers "noise" refers to amplitude noise, but it seems like is being simulated here is more like "jitter", transient noise in the phase. What is the logic in choosing "In 'noisy day start' conditions we delayed the onset of light by up to 3 h, and in 'noisy day end' conditions we delayed the transition from light to dark by up to 3 h and accelerated it by up to 1 h"? Are these numbers derived from real weather data? A premature "day end" can be understood as e.g. clouds rolling in the create a premature

dusk, but it is unclear what would physically cause a delayed "day end".

6. Worth commenting that in the Fig 6 experiments, it is possible to simulate the temporal statistics of the environment, but presumably the experimental system is not capturing the magnitude of the irradiance at these geographical sites

Reviewer #2

(Remarks to the Author)

Eremina et al. investigated the circadian clock in cyanobacteria by single-cell analysis of oscillating gene expression patterns. This study is technically sound and offers profound insights into the functioning of the *Synechococcus* circadian clock, shedding light on its robustness and underlying mechanisms.

One of the most notable strengths of this study is its innovative approach, which utilizes a microfluidic device to dissect and scrutinize the circadian rhythm at the single-cell level. This methodological choice not only enhances the precision of the analysis but also allows for a comprehensive assessment of the clock's functions with high resolution. By harnessing the power of microfluidics, researchers have effectively captured the dynamic nature of gene expression within individual cyanobacterial cells, uncovering previously obscured nuances in circadian regulation.

Because I am not a specialist in modelling, I will focus my review on single-cell analysis of wild-type and clock mutants. First, they validated the usefulness of single-cell observations by comparing oscillating reporter activity with that of previous studies using cells grown on agar. They showed that the oscillators in the single cells were very stable, and it was remarkable that they were able to show that there is a precise clock phase inheritance, which is independent of cell division. Next, they aimed to understand the role of known regulators of the cyanobacterial circadian clock on clock precision. They analyzed circadian expression in different mutant strains of clock regulators and mutations affecting KaiC function. Their model suggested that the robustness of the clock depends mainly on the robust and low-noise oscillations generated by the KaiABC oscillator. Most importantly, the authors used their single-cell based system to study simple 12 h light/ 12 h dark or more complex simulated and environmental cycles and fluctuating light conditions. These are the conditions for which one would expect clock regulators to be important in reducing noise in natural conditions. In addition to these strengths, it is important to acknowledge the limitations of this study. Despite the comprehensive analysis conducted on the wild type, a notable gap in the research lies in the absence of analyses involving clock mutants under changing environmental conditions. While this investigation provides valuable insights into the functioning of circadian clocks in a controlled setting, the exclusion of clock mutants under fluctuating conditions represents a missed opportunity to explore the full spectrum of clock robustness and adaptability. However, the establishment of single-cell analysis under controlled conditions is a significant step towards a more complete understanding of the robustness of circadian rhythms under natural conditions. Moreover, the findings presented in this study underscore the remarkable adaptability and resilience of cyanobacterial circadian clocks in response to environmental perturbations.

Specific comments:

1. Line 116: The term 'healthy cell growth and division rates' should be avoided. How "healthy" can be defined?
2. Line 269: Why were WT, KaiC-R215C, and KaiC-T495A mutants analyzed under high light, whereas SP16 and LP48 were analyzed under low light? Was there any reason for that?
3. Table 1: SynPCC 7942 is not introduced. It would be better to use the name *S. elongatus*. None of the mutants appeared to be nourseothricin resistant. Why is concentration given? What means "none" for SP16? How were regulator mutants generated? A reference is given. Did you receive strains from the respective laboratories?
4. Line 559: "To reduce phototoxicity, the maximal imaging frequency was set to every 45 min under light and every 240 min under darkness." I do not care too much about phototoxicity, but how did you exclude the possibility that excitation during microscopy can influence oscillations?

Version 1:

Reviewer comments:

Reviewer #1

(Remarks to the Author)

The authors have addressed my concerns. I think this is a valuable study and should be published.

Reviewer #2

(Remarks to the Author)

The authors have responded adequately to my minor suggestions and added the requested information on mutants. Please, check the indicated concentrations of the antibiotics. These seem to be incorrect (maybe per milliliter?). The abbreviation NSII should not be defined in Table 1, as it is not used there. Bacterial gene and mutant names should be italic (please, check especially in figures and table 1).

Response to reviewers

Environmental and molecular noise buffering by the cyanobacterial clock in individual cells

Aleksandra Eremina, Christian Schwall, Teresa Saez, Lennart Witting, Dietrich Kohlheyer, Bruno M.C. Martins, Philipp Thomas, James C. W. Locke

We thank the reviewers for their positive assessment of our manuscript. We have addressed all their points and comments in this revised version, which we detail below. We included additional experimental data sets and model simulations, and updated our figures accordingly. Specifically, the new data sets are KaiC-R215C under medium LL (Fig.3D, Fig.S3.6, S3.7), and additional data sets for WT under medium LL (Fig.S4.2), KaiC-R215C under high LL (Fig.S3.7), Δ ldpA-2 and Δ prkE-2 (Fig.S4.2). We obtained these extra data sets to address specific reviewers' points and to ensure a sufficient sample size in all experiments.

In addition, we have updated our figures and noise loop analysis to take into account a background correction using the pixel intensity of an empty channel (see Methods, lines 605-608), provided a table listing imaging conditions for all figures (Tab.S1), and clarified, in the respective figure legends, where a dataset is used in more than one figure. We note that the background subtraction resulted in the change of the C.V.² in clock amplitude (maxima of noise loops) from 0.01 to 0.05 (line 184), which is still lower than in previous reports, as we had originally concluded. All fluorescence data is now background-subtracted without any additional normalisation.

All text updates, apart from spelling corrections, are highlighted in red font. Where a significant portion of the text was removed, the text appears in red with a strikethrough line. Major figure updates are listed below:

1. Fig.1: for consistency, the same cell is highlighted in panels A-D and the same WT medium LL dataset is used throughout. Noise loops were recalculated including background subtraction.
2. Fig.2: plots in panel B modified slightly to include only lineages with consistent imaging conditions (cell counts updated accordingly). Calculations in panels C and D were not affected. Daughter phase differences in the Δ ldpA background were added to panel D (bottom).

3. Fig. 3: we chose to plot LP48 and SP16 data sets from the same chip (i.e. same experiment), so a different LP48 data set is shown (cell count adjusted accordingly) and WT is removed. Addressing reviewer 2's point on comparing the mutants under different imaging conditions, we also conducted more experiments with KaiC-R215C under medium and high LL to show that the effect of the genetics on the clock noise is similar under different light regimes. These data are now added to Fig.3D.

Updates to the supplemental figures were the following:

1. Noise calculations and noise loops in Fig.S1.5, S1.6 and S2.3 were modified to reflect background subtraction (no qualitative changes).
2. We reordered Figs.S3. Fig.S3.1 (previously Fig.S3.3) now has the autocorrelation function shown in the bottom panel; as with Fig. 3 (see above), WT medium LL was removed from Fig.S3.2-4.
3. We removed Fig.S4.2, comparing the noise loops under LD conditions, as we think it did not provide extra insight. Instead, the new Fig.S4.2 shows phase variance increases with time for all WT and regulator knock-out data sets.
4. Figs.S5 and S6 showing raw fluorescence and light data were re-ordered.

We thank the reviewers for their careful reading and suggestions, which contributed to an improved paper. Below, we address all their specific comments.

Reviewer #1 (Remarks to the Author):

There is much to like in the manuscript from Eremina et al. which uses a single cell analysis of circadian rhythms in bacteria to go beyond previous work which has touched on similar topics of stochasticity and response to external fluctuations. I see a major advance of this work to be the development of the "green mother machine" which required non-trivial technical innovation and allows for the first single cell measurements of the cyanobacterial clock under truly constant culture conditions.

We appreciate the reviewer's enthusiasm for our work, and are happy they see the development of the green mother machine as a major advance.

The study uses this mother machine tool to address multiple interconnected questions. I have some major concerns about some of the conclusions as well as more minor suggestions.

Major concerns:

1. Many readers will assume that the title "Environmental and molecular noise buffering by the core phosphorylation loop..." refers to conclusions about cells that have removed the transcriptional feedback connections between the Kai proteins and their expression. In the lingo of the field, cells that only have a "PTO". But as far as I can tell all of the experiments are on cells with intact transcriptional feedback. This seems especially relevant since the Teng et al. paper argues that the transcriptional feedback plays a crucial role in stochasticity. The title should be changed and the relationship of the analysis to previous work should be clarified.

We agree that the title could be misleading. We changed it to 'Environmental and molecular noise buffering by the cyanobacterial circadian clock in individual cells'. We also added two sentences in the discussion pointing to the role of the transcriptional feedback loop in reducing clock noise (lines 481-483).

2. The observation that two selected period mutants cause an increase in stochasticity is quite interesting, The comparison with the model in Figure 3D is questionable, since it relies on two mutants previously reported to have period effects (and the "arrhythmic mutants" where period estimation is more difficult). The model treats all period effects as arising from changes in the rates associated with

KaiA-KaiC interaction, but this seems implausible since residues 251 and 393 are quite far from KaiA interaction sites. Presumably there are other ways parameters could be changed in the model to adjust period that might produce different c.v. vs. period curves.

We apologise for the lack of clarity. We did not mean to suggest KaiA-KaiC is the only possible explanation for our observation. We have amended Fig.S3.6 to demonstrate that perturbations to several parameters in the model produce qualitatively similar behaviours. We adjusted the wording in the main text (lines 309-313) and the caption of Figure 3D to state that qualitatively similar noise minima are observed under a range of model perturbations, including changes to the dephosphorylation rate.

Perhaps another way to approach this would be to compare published (bulk) amplitudes for period mutants(see Ito-Miwa et al. for some dramatic examples)--do these agree with the Figure 3D curve?

This is an interesting suggestion. Our single-cell observations of period mutants suggest that period noise increases with deviations from the WT period. We could thus expect that period mutants display stronger damping of bulk oscillations than the WT. Indeed, the seminal Kondo et al. (1994) paper shows that bulk oscillations of both short- and long-period mutants are more damped than WT oscillations, and these include several mutations in our study. We now explicitly refer to this qualitative agreement in the main text (lines 469-472).

The study of Ito-Miwa et al. shows a wide range of periods with weak damping obtained from single residue substitutions. These mutations cannot be understood from single-parameter perturbations of our model, but other multi-parameter perturbations could be used. For example, scaling all model parameters allows tuning the period independently of the noise properties and thus leads to weak damping independent of the period. Determining exactly which parameter is changed under a specific mutation is beyond the scope of our study and remains future work. We now include a comment on and citation to this study (lines 472-476).

More minor:

3. Something seems off with Fig 3B. It seems like the difference between the top and bottom panels is rescaling of the time axis by the oscillator period. Why do the relative amplitudes of the autocorrelation peaks seem to change?

Panels A and B in Figure 3 contain data from different strains (noisy oscillator strains at the top, and period oscillator strains at the bottom). Since the mutants in panel B have non-24 clock periods, we presented autocorrelation functions normalised per period of the clock. When compared to Fig.S3.3, where autocorrelation functions of period mutants are plotted as a function of time, it can be seen that the amplitudes of the fluctuations are preserved.

4. Are the data in Fig 4B showing that phase coherence is improved at higher light at odds with data in the Teng et al. paper which seems to show increased stochasticity at higher light levels? Please comment

Teng *et al.* (2013) observed the synchronisation index dropping from 0.8 to 0.7 over the course of three days in low light and a drop from 0.7 to 0.6 over a similar timespan in high light. We observe a similar change of -0.1 in the synchronisation index over three days in our medium and low light conditions (Fig.1H). Our results are thus compatible with those observed by Teng *et al.* (2013), and this is stated in lines 174-176.

However, there are some differences between our and Teng *et al.* (2013)'s conditions, so a direct quantitative comparison is difficult. The apparent loss of synchrony at higher light levels seems to arise from imperfect entrainment of cells in Teng *et al.* (2013), which we do not observe in our microfluidic device. Furthermore, the intensity of illumination in their work was $\sim 25 \mu\text{mol m}^{-2} \text{s}^{-1}$ for high growth rate WT samples (corresponding approximately to our 'medium' light) and $\sim 10 \mu\text{mol m}^{-2} \text{s}^{-1}$ for low growth rate (corresponding to our 'low light'). We observe different growth under medium (high in Teng *et al.*(2013)) light (an average doubling time of 7-14 h in our data vs 14-16 h in their data), and even more so under low light (an average doubling time of 16-27 h in our data vs 72 h in Teng *et al.*). For these reasons, we removed the quantitative comparison.

5. The experiments in Fig 5 are quite interesting. In many papers "noise" refers to amplitude noise, but it seems like is being simulated here is more like "jitter", transient noise in the phase. What is the logic in choosing "In 'noisy day start'

conditions we delayed the onset of light by up to 3 h, and in 'noisy day end' conditions we delayed the transition from light to dark by up to 3 h and accelerated it by up to 1 h"? Are these numbers derived from real weather data? A premature "day end" can be understood as e.g. clouds rolling in the create a premature dusk, but it is unclear what would physically cause a delayed "day end".

We agree with the reviewer that only dawn delay and dusk advance are representative of the conditions that cyanobacteria can encounter in the wild (although some organisms could experience delays in dusk due to artificial human-made light pollution). Overall, the experiments in Fig.5 were conceptual and aimed at probing the clock's resilience to perturbations in the timing of L/D transitions, both expected (or 'natural') in the wild and unexpected (or 'artificial'). We now clearly state in the main text that the perturbations in Fig. 5 probe the response to artificial environmental noise while Fig. 6 concerns real environmental conditions (lines 355-357, 395-396).

6. Worth commenting that in the Fig 6 experiments, it is possible to simulate the temporal statistics of the environment, but presumably the experimental system is not capturing the magnitude of the irradiance at these geographical sites

The light conditions used in the experiments correspond directly to the meteorological measurements by Medina et al. (2018) 3 m underwater. Overall, the irradiance magnitude on the site varied from $\sim 10 \mu\text{mol m}^{-2} \text{s}^{-1}$ to $\sim 300 \mu\text{mol m}^{-2} \text{s}^{-1}$. We selected the sequence of days closest to the conditions in Figures 1-4 in their cumulative light intensities. While these are the least bright days of the year, the conditions we used are still representative of the light irradiance on the site.

We have added clarification that the experiment and the parametrised model represent realistic irradiances. This is included in the second to last paragraph of the Results and caption of Fig.6A.

Reviewer #2 (Remarks to the Author):

Eremina et al. investigated the circadian clock in cyanobacteria by single-cell analysis of oscillating gene expression patterns. This study is technically sound and offers profound insights into the functioning of the Synechococcus circadian clock, shedding light on its robustness and underlying mechanisms.

One of the most notable strengths of this study is its innovative approach, which utilizes a microfluidic device to dissect and scrutinize the circadian rhythm at the single-cell level. This methodological choice not only enhances the precision of the analysis but also allows for a comprehensive assessment of the clock's functions with high resolution. By harnessing the power of microfluidics, researchers have effectively captured the dynamic nature of gene expression within individual cyanobacterial cells, uncovering previously obscured nuances in circadian regulation.

We are grateful for the reviewer's positive assessment of our work. We address their comments below, point by point:

Because I am not a specialist in modelling, I will focus my review on single-cell analysis of wild-type and clock mutants. First, they validated the usefulness of single-cell observations by comparing oscillating reporter activity with that of previous studies using cells grown on agar. They showed that the oscillators in the single cells were very stable, and it was remarkable that they were able to show that there is a precise clock phase inheritance, which is independent of cell division. Next, they aimed to understand the role of known regulators of the cyanobacterial circadian clock on clock precision. They analyzed circadian expression in different mutant strains of clock regulators and mutations affecting KaiC function. Their model suggested that the robustness of the clock depends mainly on the robust and low-noise oscillations generated by the KaiABC oscillator. Most importantly, the authors used their single-cell based system to study simple 12 h light/ 12 h dark or more complex simulated and environmental cycles and fluctuating light conditions. These are the conditions for which one would expect clock regulators to be important in reducing noise in natural conditions. In addition to these strengths, it is important to acknowledge the limitations of this study. Despite the comprehensive analysis conducted on the wild type, a notable gap in the research lies in the absence of analyses involving clock mutants under changing environmental conditions. While this investigation provides valuable insights into the functioning of circadian clocks in a controlled setting, the exclusion of clock mutants under fluctuating conditions represents a missed opportunity to explore the full spectrum of clock robustness and adaptability. However, the establishment of single-cell analysis under controlled conditions is a significant step towards a more complete understanding of the robustness of circadian rhythms under natural conditions. Moreover, the findings presented in this study underscore the remarkable adaptability and resilience of cyanobacterial circadian clocks in response to environmental perturbations.

We agree that examining clock regulator mutants under fluctuating environments would be fascinating, and although out of the scope of this study, it is something we intend to do in future work. We now point to this open question in the fifth paragraph of the discussion (lines 500-502).

Specific comments:

1. Line 116: The term 'healthy cell growth and division rates' should be avoided. How "healthy" can be defined?

We agree, and have omitted the word 'healthy' and simply state agreement with phenotypes observed on agarose pads.

2. Line 269: Why were WT, KaiC-R215C, and KaiC-T495A mutants analyzed under high light, whereas SP16 and LP48 were analyzed under low light? Was there any reason for that?

There was no specific reason for this. We added new experiments shown in Fig.S3.7 that demonstrate KaiC-R215C clock dynamics are similar under both medium and high LL. Moreover, Fig.3D shows that the differences in clock period noise due to light levels are negligible compared with genetic effects.

3. Table 1: SynPCC 7942 is not introduced. It would be better to use the name *S. elongatus*. None of the mutants appeared to be nourseothricin resistant. Why is concentration given? What means "none" for SP16? How were regulator mutants generated? A reference is given. Did you receive strains from the respective laboratories?

We apologise for the lack of clarity. The mutations were generated in our lab and no nourseothricin was used in generating the mutants. We have now corrected this point and have included extra details in Table 1.

4. Line 559: "To reduce phototoxicity, the maximal imaging frequency was set to every 45 min under light and every 240 min under darkness." I do not care too much about phototoxicity, but how did you exclude the possibility that excitation during microscopy can influence oscillations?

Fluorescence microscopy imaging of clock reporter strains in cyanobacteria is an established technique (reviewed for example in Cohen SE et al. (2015); and Yokoo et al. (2015)). Common imaging practices that we use to reduce the effects of excitation

light or other artefacts are: (i) a low imaging frequency (> 30 min); (ii) short exposure time (≤ 0.12 s with a highly localised light cone, which amounts to a very short exposure when compared to the light cues with time scales of dozens of minutes or hours that cause measurable phase shifts); and (iii) avoiding fluorescent reporters with excitation spectra overlapping peaks of cellular autofluorescence (blue and red wavelength windows mainly). We note that similar growth rates were observed without excitation illumination and in strains without fluorescent reporters grown on agarose pads (see Martins *et al.* (2018)).

Reviewer #2 (Remarks to the Author):

The authors have responded adequately to my minor suggestions and added the requested information on mutants. Please, check the indicated concentrations of the antibiotics. These seem to be incorrect (maybe per milliliter?). The abbreviation NSII should not be defined in Table 1, as it is not used there. Bacterial gene and mutant names should be italic (please, check especially in figures and table 1).

Thank you for these corrections. We have fixed the antibiotic concentrations and italicised gene names and knockouts. NSII is used as a heading in the Table, so we have retained the definition in the legend.